# Evaluation of Scab and Mildew Resistance in the Gene Bank Collection of Apples in Dresden-Pillnitz

**DOI:** 10.3390/plants10061227

**Published:** 2021-06-16

**Authors:** Monika Höfer, Henryk Flachowsky, Susan Schröpfer, Andreas Peil

**Affiliations:** Julius Kühn Institute (JKI)—Federal Research Centre for Cultivated Plants, Institute for Breeding Research on Fruit Crops, Pillnitzer Platz 3a, 01326 Dresden, Germany; henryk.flachowsky@julius-kuehn.de (H.F.); susan.schroepfer@julius-kuehn.de (S.S.); andreas.peil@julius-kuehn.de (A.P.)

**Keywords:** apple, biodiversity, disease resistance, fruit tree resources, mildew, scab, SSR marker

## Abstract

A set of 680 apple cultivars from the Fruit Gene bank in Dresden Pillnitz was evaluated for the incidence of powdery mildew and scab in two consecutive years. The incidence of both scab and powdery mildew increased significantly in the second year. Sixty and 43 cultivars with very low incidence in both years of scab and powdery mildew, respectively, were analysed with molecular markers linked to known resistance genes. Thirty-five cultivars were identified to express alleles or combinations of alleles linked to *Rvi2*, *Rvi4*, *Rvi6*, *Rvi13*, *Rvi14*, or *Rvi17*. Twenty of them, modern as well as a few traditional cultivars known before the introduction or *Rvi6* from *Malus floribunda* 821, amplified the 159 bp fragment of marker CH_Vf1 that is linked to *Rvi6*. Alleles linked to *Pl1, Pld*, or *Plm* were expressed from five cultivars resistant to powdery mildew. Eleven cultivars were identified to have very low susceptibility to both powdery mildew and scab. The information on resistance/susceptibility of fruit genetic resources towards economically important diseases is important for breeding and for replanting traditional cultivars. Furthermore, our work provides a well-defined basis for the discovery of undescribed, new scab, and powdery mildew resistance.

## 1. Introduction

Global climate change and environmental degradation increasingly endanger many areas of our life. New concepts are required to counteract these ongoing processes. The European Green Deal is the new European growth strategy and a roadmap to make Europe climate-neutral until the middle of this century. The Farm to Fork Strategy (https://ec.europa.eu/food/farm2fork_en, accessed on 29 October 2020) is an important part of the Green Deal. As current food production systems are key drivers of climate change and environmental degradation, the Farm to Fork strategy recognizes, among others, the urgent need to reduce the dependency on pesticides and excess fertilisation, to increase organic farming, and to fight against the progressive loss of biodiversity.

The rising incidence of biotic and abiotic stresses makes commercial fruit production increasingly difficult in Europe. Especially orchards affected by fungal diseases like apple scab and apple powdery mildew, which are caused by *Venturia inaequalis* (Cooke Winter, 1875) and *Podospaera leucotricha* (Ellis & Everh. Salmon), respectively, require intensive plant protection management. In integrated fruit production, up to 20 fungicide treatments (equalling to a fungicide treatment index of up to 26) are applied [1] to avoid severe scab and powdery mildew infections. Organic management practices, which are known to have a higher output of ecosystem services, are associated with a 48% lower yield compared to integrated fruit production [2]. Planting resistant cultivars is a promising strategy. Some resistant apple cultivars with sufficient fruit quality are already available. Most of them carry the *Rvi6* scab resistance gene originating from *Malus floribunda* Siebold ex. Van Houtte clone 821. Unfortunately, *Rvi6* has already been overcome in several European fruit-growing regions [3], but also in other parts of the world [4]. Top cultivars with durable resistance and excellent fruit quality are urgently required to make future apple fruit production more eco-friendly, robust and resilient. During the last decades of the 20th century, breeding objectives have mainly focused on meeting aesthetic standards, with eating quality and durable disease resistance receiving greater priority [5]. Many breeding programs worldwide are meanwhile aiming on breeding for durable disease resistance towards apple scab and powdery mildew, often combined with reduced susceptibility to other diseases (e.g., fire blight, cancer) and pests (e.g., insects), but also on tolerance to abiotic stresses like frost, heat, drought, and UV radiation [6]. However, improving the host resistance to apple scab and powdery mildew is mostly the major goal [6,7]. In this context, combining different resistance (R) genes in the same genotype (pyramiding) is considered a reliable way to achieve more durable resistance [8]. For apple scab resistance, roughly 20 *R genes* are already mapped in different genetic backgrounds [7]. Some of them, like *Rvi1*, *Rvi3*, *Rvi8*, and *Rvi10*, have only little value for breeding since scab strains that are able to overcome these genes are widely distributed in Europe [3]. Other *R* genes, such as *Rvi2*, *Rvi4*, *Rvi6*, *Rvi7*, *Rvi9*, and *Rvi13*, are only useful if used in combination with at least two other scab resistance genes. The only sources for durable resistance, which are currently available, are *Rvi5*, *Rvi11*, *Rvi12*, *Rvi14*, and *Rvi15* [3]. Four of them originated from small-fruited wild apple accessions and need to be introgressed into elite material. Although, about 20 *R* genes are available, the possibilities for scab resistance breeding are still restricted. The genetic base for improving the resistance to powdery mildew is even more limited [9]. Much less is known about suitable sources for improving other traits, which are expected to become affected by the climate change. Broadening the genetic base is therefore necessary to meet the future challenges in apple breeding [10,11].

Progress in fruit breeding strongly depends on the availability of a rich diversity of genetic resources. In Germany, the conservation of fruit varieties dates back to the early decades of the 20th century, and cultivars of different fruit crop species are now preserved in public and private germplasm collections. The German Fruit Gene bank (GFG—https://www.deutsche-genbank-obst.de, accessed on 10 May 2021) was recently established as a decentralized network to coordinate the activities of the different germplasm collections [12]. The apple network which is part of the GFG was founded in 2009 and comprises 745 apple cultivars with a GFG mandate held by 14 stakeholders [13]. The main partner of the apple network, the Julius Kühn Institute (JKI), Institute for Breeding Research on Fruit Crops in Dresden-Pillnitz, maintains an apple gene bank with 702 cultivars, consisting of mostly old German cultivars or cultivars with a socio-cultural, local and historical relation to Germany [14]. The Fruit Gene Bank at the JKI, Institute for Breeding Research on Fruit Crops has been established as a partner for science and practice at the national and international level since its integration into JKI at 1 January 2003. Since 2003, the Fruit Gene bank has focused its activities on the maintenance and restructuring of the collections as well as on the evaluation of existing plant material. The Fruit Gene bank focuses on fruit species native to Central Europe and species which are important for fruit production in Germany in the present as well as in the past. These genetic resources will be comprehensively evaluated for a multitude of traits, which are of interest for breeding. Interesting genotypes will be continuously selected. After estimating their breeding value, they will be directly subjected to the breeding program.

On this basis, this work makes a decisive contribution to the implementation of the ‘‘National Program for Genetic Resources of Agricultural and Horticultural Plants’’ in Germany, whose aim is to develop an effective conservation strategy for fruit genetic resources, and to evaluate these resources in order to use them in fruit production, breeding and research.

The present study was aimed at the evaluation of genetic resources of the apple (*Malus domestica* Borkh.) cultivar collection of the Fruit Gene bank in Dresden-Pillnitz, Germany for their resistance to apple scab and apple powdery mildew and the detection of hitherto unknown resistances to these diseases. Therefore, cultivars showing resistance to scab or powdery mildew in two consecutive years were screened with a set of molecular markers for the putative presence of scab and/or powdery mildew resistance genes. Resistant cultivars not showing marker alleles linked to known resistance genes were selected as new potential sources for resistance breeding. Phenotypic results were compared to data available from previous experiments, which were performed in 1997, 1999, 2006, and 2007.

## 2. Material and Methods

### 2.1. Plant Material

Six hundred and eighty apple cultivars, which belong to the gene bank collection of the JKI Institute for Breeding Research on Fruit Crops (Dresden-Pillnitz, Germany), were used in this study (Appendix A). The institute is located at 51°0000700 N latitude, and 13°5205900 E longitude, altitude 115 m, with 9.1 °C annual average temperature and 668 mm annual precipitation. The soil type at the orchard is clayey sand with pH 5.6–6.6. Trees of these cultivars were grown on M9/Hibernal rootstock/interstock combinations with two replicates per genotype. The trees are between 4 and 8 years old and were trained as ‘Spindelbusch’. Trueness-to-type was assured in a two-step procedure consisting of a pomological characterization (step 1) and a DNA-fingerprint analysis (step 2) using a set of SSR markers suggested by the European Cooperative Programme for Plant Genetic Resources (ECPGR, Höfer, Flachowsky and Hanke [13]). At least two experts, preferably members of the German Pomological Society, performed the pomological characterization. Plant protection was carried out in accordance to integrated fruit production. In years where trees were evaluated on fungal disease symptoms (1997, 1999, 2006, 2007, 2012, 2013) no fungicides were applied.

Selected genotypes of the scab differential host set (http://www.vinquest.ch/monitoring/establishing_network.htm, accessed on 10 May 2021), complemented by four genotypes containing the apple scab resistance gene *Rvi17* (04-214-79), the powdery mildew resistance genes *Plm* (Mildew Immune seedling—MIS) and *Pld* (D12), or a combination of the three mildew resistance genes *Pl1*, *Pl2*, and *Plm* (06_57) were used as control.

### 2.2. Phenotyping of Powdery Mildew and Scab

Field evaluation on the occurrence of apple scab (*V. inaequalis*) and apple powdery mildew (*P. leucotricha*) symptoms on leaf laminae was conducted twice a year (June and August) and once for fruit scab (August–September) in 2012 and 2013. Scoring of scab and powdery mildew symptoms was performed according to the assessment scale defined in Table 1.

Phenotypic data recorded in 1997, 1999, 2006, and 2007 were available from previous experiments (Table 2 and Table 3). Data recorded in 1997 and 1999 have been partly published by Fischer and Dunemann [15], as well as Fischer and Fischer [16] already. In the present study, data were recorded in the same orchard in 2006, 2007, 2012, and 2013.

### 2.3. Molecular Analysis

Cultivars with leaf and fruit scab and powdery mildew ratings, respectively, lower than 3 in both years (2012 and 2013) were analysed using molecular markers for scab and/or powdery mildew resistance genes (Appendix A) [17,18,19,20]. Sixty-two cultivars were screened with markers for scab resistance genes, whereas 43 cultivars were checked for the presence of markers for resistance to powdery mildew.

DNA was extracted using the REDExtract-N-Amp^TM^ Plant Kit (Sigma-Aldrich, Darmstadt, Germany). Leaf discs of 3 mm diameter were incubated in 50 µL extraction solution for 10 min at 95 °C and finally 50 µL dilution buffer was added. DNA was diluted 1:5 with water and 1 µL was used for PCR. DNA of the control genotypes was isolated from young leaves using the Plant Mini Kit (Qiagen, Hilden, Germany) according to the manufacturer’s instructions. DNA was diluted to a final concentration of 10 ng µL^−1^ and 1 µL DNA was used for PCR.

Amplification of resistance markers was done in multiplexes (MPs) with fluorescence labelled forward primers (Appendix A). PCR was performed using the Type-It Kit (Qiagen) according to the manufacturer’s protocol, but in a volume of 6 µL. After initial denaturation at 95 °C for 5 min 30 PCR cycles were performed consisting of 95 °C for 1 min, 60 °C for 1 min 30 s, and 72 °C for 30 s. After a final extension for 30 min at 60 °C the PCR fragments were analysed on an Applied Biosystems 3500xL Genetic Analyser (Thermo Fisher, Berlin, Germany). Therefore, 195 µL water was added to each PCR sample, 1 µL of diluted PCR product was added to 8.95 µL HiDi and 0.05 µL Gene Scan^TM^ 600 LIZ^TM^ Dye Size Standard (ThermoFisher). Fragment analysis was done using the Applied Biosystems^TM^ GeneMapper™ 6 software (ThermoFisher). Sizes of fragments linked to resistance genes were compared to fragments amplified by the control genotypes. Allele sizes were adjusted to published allele sizes and/or allele sizes produced by Gala and Golden Delicious used as additional controls. The multiplexes MP1 to MP6 were applied to cultivars listed in Table 2 and MP1_mild to cultivars listed in Table 3. SSR markers Ch02d12, Ch03c02, Ch04h02, and SSR-23.03 were additionally applied as single markers.

### 2.4. Development of Markers for the Genes Pl2 and Rvi15

Sequences for the development of gene specific primers for *Pl2* and *Rvi15* were obtained from the United States patent (No. US20100306875A1) and the Gene bank accession number KF055410 (Schouten et al. [21]), respectively. Primer pairs were designed using Primer3 and their specificity was tested on selected individuals with and without *Pl2* and *Rvi15* (unpublished data), respectively. Primer pairs Pl2_F1/R1 (Pl2_F1 GTGGTGTTTTCCCTTTCCTA; Pl2_R1 TCCTACTATGCGAAGCTTTT) and Vr2C_UTR (Vr2C5_UTR_F GTTTCTTAGGAAGGGATATAGGGCAGCA; Vr2C5_UTR_R GTTTCTTCCAACTCGCGAATTTAGCCG), specific for the respective genes *Pl2* and *Rvi15*, were chosen for marker analysis.

### 2.5. Data Analysis

Statistical analyses were performed using SAS Enterprise Guide 4.3 or SAS 9.4. (SAS Institute Inc.). Frequencies of resistance classes were calculated for powdery mildew and scab symptoms on leafs and fruits. Only genotypes for which data from both experimental years (2012 and 2013) were available were included (Figure 1). For scab infestation of fruit and leaf, data of 539 and 629 cultivars, respectively, were analysed. For powdery mildew infestation, data of 632 cultivars were analysed. The SAS GLIMMIX (Generalized Linear Mixed Model) procedure [22] was performed to determine whether the trial year 2012 and 2013 caused significant differences in the average score, at a significance level of α = 0.05.

## 3. Results

### 3.1. Field Evaluation of Scab and Mildew Susceptibility

Detailed data for each genotype are shown in Appendix A. Ninety-four (14.9%) out of 629 cultivars showed no scab symptoms on leaves or only a few small spots (scales 1 and 2) in 2012 and 2013. For fruit scab, this number was higher with 147 out of 539 cultivars (27.3%). Levels of leaf scab of 8 or 9 at least in one year were found for 6.7% of the cultivars. The percentage of cultivars with high levels of fruit scab was around three times higher (22.4%). A correlation between leaf and fruit scab incidence was found with r = 0.67 and r = 0.66 for 2012 and 2013, respectively. The mean incidence for leaf and fruit scab in 2013 with values of 4.7 and 4.6 was significantly higher than the values of 2.6 and 1.6 observed in 2012. Five-hundred and one and 389 cultivars showed a higher incidence of leaf and fruit scab in 2013 than in 2012, respectively, and 41 and 10 cultivars a higher incidence of leaf and fruit scab in 2012 than in 2013. Figure 1 shows the progress of leaf and fruit scab in 2013 compared to 2012. Sixty cultivars showed ratings lower than 3 in 2012 and 2013 for both leaf and fruit scab and were subsequently analysed with molecular markers.

The group of cultivars with no or very few scab symptoms consisted of traditional cultivars, but also of newly breed resistant cultivars such as Ariwa, Realka and Reka. The group of traditional cultivars consisted of widely distributed cultivars including Antonovka, Finkenwerder Prinzenapfel and Prinz Albrecht von Preußen, as well as local cultivars, such as Börtlinger Weinapfel, Linsenhofer Sämling, and Oetwiller Renette, for example. The current gene bank collection resulted from replanting of a previous collection initially grown at the same place. For this previous collection, data from scab evaluation were already available. For 42 accessions of scab resistant cultivars, data of multiple years were available (Table 2). Interestingly, some cultivars with no or very low scab symptoms in some years (e.g., Oberöstereichischer Brünnerling and Retina) showed a higher incidence in other years with values of up to 4. Five cultivars (Remura, Reka, Nela, Engelshofer, and Altländer Rosenapfel) were without scab symptoms in all three periods, altogether 26 cultivars showed always rating scales of ≤2.0.

Higher incidence was observed for powdery mildew. Only 43 (6.8%) out of 632 cultivars showed very low susceptibility to powdery mildew (scales 1 and 2) in 2012 and 2013 (Table 3). Figure 1 shows the distribution of scab ratings in 2012 and 2013. The mean incidence of mildew was significantly higher in 2013 than in 2012 with ratings of 3.7 and 4.3, respectively. Three hundred thirty-eight cultivars showed a higher mildew incidence in 2013 compared to 2012 and 126 cultivars had higher incidence in 2012 than in 2013. Data from previous experiments for cultivars were available for 21 accessions (Table 3). The incidence to mildew showed a greater variability for previously mentioned 42 cultivars over the three periods compared to scab, with values of up to 7 for e.g., Wildeshausener Goldrenette in 2006/2007. No cultivar was found which was free of mildew symptoms in all three periods. However, mean rating scales of ≤2.0 were detected for Peasgoods Sondergleichen, Kardinal Bea, Erbachhofer, and Baumanns Renette (no data available for 1997 and 1999).

The comparison of scab and powdery mildew ratings lead to the identification of 11 varieties, which exhibit low susceptibility to both diseases at least in 2012 and 2013 (Table 2 and Table 3). Among them, there are modern varieties like Ariwa and Realka, but also traditional ones like Altländer Rosenapfel, Börtlinger Weinapfel or Porzenapfel.

### 3.2. Validation of Marker Alleles Linked to R Genes Using a Set of Selected Apple Genotypes

The 60 cultivars showing scab ratings lower than 3 in 2012 and 2013 for both leaf and fruit scab and the 42 cultivars with mildew ratings lower than 3 in both years were subjected to marker analysis. Markers were first tested using a set of selected genotypes used as controls to validate the size of alleles linked to *R* genes (Table 4), and then subsequently analyzed with the chosen cultivars. Most marker fragments amplified in this study (Table 4) differed in size by a few base pairs compared to published data (Table 4). However, the size difference between the individual alleles of a marker reported in literature were in general the same in this study. More than one marker was applied for some *R* genes, and some markers are coupled with different *R* genes (Table 4). The rationale of the analyses for the respective genes are explained below (no markers were available for *Rvi7*, *Rvi9*, and *Rvi10*), detailed data and references are given in Table 4.

**Rvi1:** VG12_SSR and VG15_SSR were used as markers for Rvi1. Since no information about the alleles linked to Rvi1 are available, no data are presented for these markers. **Rvi2**: Markers CH02b10, CH05e03, and OPL19SCAR were used to determine putative presence of Rvi2. The presence of Rvi2 was assigned only if a cultivar amplified the alleles linked to Rvi2 for all three markers. Marker OPL19SCAR amplified several unspecific bands of the same size as the allele linked to Rvi2. **Rvi3**: No information about the fragment size linked to resistance of Hi08e04 is available from literature. Therefore, no data are presented for this marker. **Rvi4**, **Rvi15**: The 176 bp allele of marker CH02c02a indicates the presence of *Rvi4* as well as the presence of *Rvi15*. Patocchi et al. [23] observed allele sizes of 152 bp (linked to *Rvi15*) and 165 bp for CH02f06, when applied to the differential scab host H(15). In the present study, 147 bp and 159 bp were observed. The 147 bp allele, which was determined as linked to *Rvi15*, was additionally amplified in the scab differential hosts H(2), H(4), H(6), and H(9). Vr2C_UTR, which was developed from the sequence of *Rvi15***,** amplified the respective fragment only in H(4) and H(15). Since *Rvi4* and *Rvi15* could not be distinguished by all three markers, both genes were assigned to cultivars that possessed alleles linked to resistance of all three markers, if the pedigree of a cultivar could not help to discriminate between both genes. **Rvi5**: If a cultivar exhibits the alleles linked to resistance of all three markers, Hi07h02, FMACH_Vm2, and FMACH_Vm3, the presence of Rvi5 was assumed. **Rvi6**, **Rvi17**: The presence of Rvi6 and/or Rvi17 was assigned if a cultivar possesses the 159 and/or 139 bp allele with CH_Vf1, respectively, linked to resistance. **Rvi8**: Markers OPB18SCAR and OPL19SCAR are indicators for *Rvi8*. Bus et al. [24] reported an allele size of 799 bp linked to *Rvi8* for marker OPB18SCAR. In the present study, a fragment of approximately 755 bp was detected, when applied to H(8). However, both fragment sizes are out of the range of the size standard (all other cultivars amplified fragments smaller than 700 bp). Therefore, the observed fragment size is just an estimation but was used together with the 430 bp of OPL19SCAR to assign the presence of *Rvi8*. ***Rvi11***: Differences in size were observed for markers CH03d01 and CH02c06 (linked to *Rvi11*). The fragments obtained for H(11), i.e., *M. baccata* var. jackii, did not match the allele expected sizes published by Gygax et al. [25]. Only the 150 bp fragment of marker CH05e03 determines the linkage to *Rvi11*. Patocchi et al. [23] detected a 160 bp fragment with linkage to *Rvi11* using this marker. However, they also detected 10 to 11 bp longer fragments for Gala and Golden Delicious (Table 4). No allele with the supposed fragment size of 410 bp for linkage of the T6 marker to *Rvi11* was detectable. All observed fragments had sizes larger than 600 bp. ***Rvi12***: Marker alleles of SSR-23–17 and SSR-24.91 showed 1–2 bp differences in size compared to the reference. Both alleles linked to Rvi12 were found only in H (12). ***Rvi13***: Both the 111 bp and 185 bp alleles of markers CH02b07 and CH04f03 indicate the presence of *Rvi13*. ***Rvi14***: The 210 bp fragment of HB09 indicates *Rvi14.* ***Rvi16***: Genotype MIS was used to determine the allele sizes linked to *Rvi16* amplified under the conditions applied at JKI by NZmsCN943818 and NH030a. NZmsCN943818 and NH030a showed differences of 19 and 17 bp in allele size, respectively, compared to the findings of Bus et al. [26]. The presence of *Rvi16*, showing both alleles linked to resistance, was observed for MIS only.

***Pl1***: The 450 bp allele amplified by AT20Scar indicates the presence of *Pl1*. ***Pl2***: The 573 bp allele of Pl2_F1/R1 indicates the presence of *Pl2*. Ch04h02 as single was screened only on the 42 cultivars with scab incidences lower than 3 in 2012 and 2013. The 180 bp fragment indicating *Pl2* was detected in D12 and the *Pl1 Pl2 Plm* reference 06_57 only (data not given in Table 4). ***Pld***: A difference in allele size distance was also observed for marker CH03c02. In contrast to James et al. [27], who reported allele sizes of 125 bp and 131 bp for D12, allele sizes of 106 bp and 129 bp were detected in the present study. In comparison with the allele sizes the same authors obtained for MIS (Table 4), which indicates a small size shift of 2–3 bp, the 129 bp fragment amplified by D12 (this study) was assigned as marker for *Pld*. ***Plm***: A difference of 20 bp was observed for marker CH02d12 linked to *Plm for both alleles.* Bus et al. [26] reported allele sizes of 197 and 205 bp (coupled to *Plm*), when applied to MIS. Our analysis revealed fragment sizes of 177 and 185 bp. The 185 bp fragment was assigned as the R gene specific allele.

### 3.3. Evaluation of Resistant Cultivars for the Presence of Marker Alleles Linked to Known R Genes

The marker linked to *Rvi6* was detected in 20 cultivars (Table 2). Among them, there were newly bred cultivars like Ariwa and Topaz, but also traditional ones like Früher Viktoria, Grenadier, Odenwälder, and Purpurroter Cousinot (Table 2). Alleles linked to both *Rvi6* and *Rvi13* were detected in four cultivars. Generos, Jeverländer Süßapfel (pomologically not determinable), and Schöner aus Elmpt amplified marker alleles linked to *Rvi13*. The HB09 allele of 210 bp linked to *Rvi14* was detected in Altländer Rosenapfel, Hadelner Sommerprinz, and Hochzeitsapfel. The cultivars Antonovka, Antonovka Kamenička, and Hibernal amplified marker alleles linked to *Rvi14* and *Rvi17*. For Angold and Bessemânka Mičurinskaâ only the CH-Vf1 allele of 139 bp linked to *Rvi17* was detected. A combination of marker alleles linked to the three scab resistance genes *Rvi2*, *Rvi4* and *Rvi6* were found for Realka. From Reka, alleles specific for *Rvi2* were amplified, whereas in Remura alleles for *Rvi4* were detectable. Six cultivars with rating scales for scab of ≤2.0 in all three periods amplified none of the alleles linked to resistance of all markers tested.

Marker alleles linked to the powdery mildew resistance genes *Pl1*, *Pld*, and *Plm* were found in only a few cultivars (Table 3). The Swiss cultivar Ariwa possesses *Rvi6* and *Pl1* (Table 2 and Table 3).

## 4. Discussion

Breeding and cultivation of disease-resistant apple varieties is a forward-looking strategy that aims to reduce the use of pesticides and may contribute to a sustainable agricultural economy. The evaluation and characterization of available genetic resources of a cultivated species with regard to resistance against diseases represents a great opportunity to identify resistant genotypes and furthermore, to use biodiversity to counteract existing challenges in the production of fruits. The major problems in apple production are caused by fungal pathogens like *V. inaequalis* and *P. leucotricha,* causing scab and powdery mildew respectively.

Within the framework of this study, cultivars with resistance to scab and/or powdery mildew could be identified in the apple collection of the Fruit Gene bank in Dresden-Pillnitz. At the time of the evaluation, the apple collection consisted of 702 different cultivars, containing mainly old German cultivars or cultivars with a socio-cultural, local, and historical relation to Germany [14]. The obtained data extend previous work on the evaluation for disease resistance of collections from other European gene banks, for example in Belgium, Sweden and Switzerland [33,34,35,36,37]. The classification of cultivars collected within gene banks in terms of their susceptibility to diseases is usually performed under field conditions. This needs to be included in consideration, as many location- and year-specific factors can greatly influence the evaluation results, e.g., year-to-year variation is highly dependent on inoculum pressure. Furthermore, the evaluation period is typically preceded by a period of standard fungicide treatment, which is necessary to preserve the collection of the gene bank. Therefore, it takes several years before disease inoculum become fully established in unsprayed orchards and also the spatial distribution can vary greatly in the same orchard [37]. In order to achieve reliable results, field evaluation data from several years should to be considered. This was accomplished in the present study by analyzing data obtained from the same orchard in the evaluation period 2012–2013 (this work), 1997, and 1999, as well in the period from 2006 to 2007 (partly published by Fischer and Dunemann [15], Fischer and Fischer [16]), which were available for most of the analyzed cultivars.

The observed span of susceptibility responses to scab and mildew was high (Figure 1), ranging from cultivars exhibiting no symptoms to a maximum of infection, which indicates the presence of effective disease inoculum during the evaluation period. From cultivars classified as scab resistant in 2012–2013, five cultivars were without scab symptoms in all regarded evaluation periods and additional 22 cultivars could be identified showing no or very low susceptibility. The presence of modern cultivars in this group of resistant cultivars, e.g., Reka, Remura, Realka, demonstrate the success of scab resistance breeding programs. Compared to scab evaluation data, the year-to-year variance was clearly higher for powdery mildew. This might explain the finding that only 43 cultivars were identified as resistant to powdery mildew in the evaluation period 2012–2013, and only four cultivars exhibit no or very low susceptibility in all considered periods. A combination of low susceptibility to both diseases was found in 11 cultivars. In addition to modern cultivars originating from resistance breeding programs, there are also traditional varieties that have good resistance properties. While modern varieties are extensively described in literature [16,38], traditional varieties are only occasionally mentioned in previous investigations in relation to scab [15,39]. Some of these older cultivars exhibiting a low susceptibility to powdery mildew, such as Baumanns Renette, Peasgoods Sondergleichen, Gaesdonker Renette, and Altländer Pfannkuchenapfel, have already been considered in mildew evaluation studies from the early and middle of the 20th century [40,41,42].

Resistance to disease is based on genetically fixed traits. For breeding of resistant varieties, in addition to the known phenotypic resistance response of the starting material, the genetic traits are also of crucial importance. Several different resistance traits, so-called *R* genes, are described for scab, and some are known for powdery mildew. However, the type, quality, and durability of the resistance conferred by the *R* genes varies. For the identification of *R* genes, molecular markers are applied in some breeding programs to select for resistant seedlings [43] and they are also used for genotyping of apple genetic resources [44,45]. Nevertheless, the correct assignment of marker alleles coupled to resistance is often very difficult if the same genotype used to develop the marker is not simultaneously analyzed [23]. Furthermore, the comparability of measured allele fragment sizes from independent studies becomes more difficult, e.g., by the use of different lab equipment or fluorescent dyes. In this work, the determination of marker alleles coupled to resistances was performed according to Patocchi et al. [23], who presented an approach for the standardization of fragment sizes of alleles linked to scab resistance. Furthermore, genotypes with known *R* genes were used for the identification of marker alleles linked to resistance. However, the use of molecular markers that are mostly located in spatial proximity to the locus of the R gene should be considered as an indication of the presence of the R gene only, since recombination events may disconnect both genomic loci. The closer the locus of marker and the R gene are, the lower the probability of recombination between them, thus increasing the informative value of the marker [6]. The most reliable results will be obtained by the detection of the R gene itself. In this work, such markers were developed for the detection of *Pl2* (*Pl2_F1/R1*) and *Rvi15* (Vr2C_UTR) specific gene sequences.

Of all the *Rvi* genes described to date, *Rvi5*, *Rvi11*, *Rvi12*, *Rvi14*, and *Rvi15* confer durable resistance to scab [3] and therefore they are of special interest for resistance breeding. From these genes, only *Rvi14* was detected in a total of seven cultivars of the gene bank collection classified as scab resistant. In four old traditional cultivars (Altländer Rosenapfel, Englischer Prinz, Hadelner Sommerprinz, Hochzeitsapfel) *Rvi14* was detected as single a R gene, and three further genotypes originating in the eastern of Europe possess a combination of *Rvi14* and *Rvi17* (Antonovka, Antonovka Kamenička, Hibernal 4n). From today’s perspective, these cultivars represent a genetic source for a durable resistance to scab.

The most abundant R gene among the scab resistant cultivars from the gene bank was *Rvi6*. This finding is according to the expectation, because the scab resistance mediated by *Rvi6* was often used in modern resistance breeding programs and can be traced back to crosses done in 1914 and 1915 by Crandall [46], who was using the resistant wild species *M. floribunda* 821 as donator for resistance [47]. Until now, most of the scab resistant cultivars released carry only *Rvi6* [3]. However, the value of resistance mediated by *Rvi6* is weakened by the occurrence of *avrRvi6* races of the pathogen in Europe [48,49,50] and in the US [51], which are able to break the resistance of *Rvi6*. This fact clearly demonstrates that the use of single *R* genes for durable resistance is not effective in the long term, and suggests a combination of different *R* genes for new breeds, as it is done in pyramidization breeding programs. Interestingly, markers linked to *Rvi6* were also detected in older cultivars originating before the initial crossing with *M. floribunda* 821 was described. Examples for those cultivars are Früher Victoria (described in 1899), Purpurroter Cousinot (around ~1600), and Grenadier (1862). A second sampling and R gene determination confirmed these results. Molecular fingerprinting and pomological analysis (unpublished data) demonstrated trueness-to-type of these cultivars.

While the durability and effectiveness of *Rvi* genes has been observed for a long period [3], little information is known in this regard for the *R* genes mediating resistance to powdery mildew. For a large proportion of scab and powdery mildew resistant cultivars analyzed in this study, none of the previously described *R* genes could be determined. These genotypes are also of great interest, as they could represent sources of unknown resistance traits. Therefore, the aim of further work should focus on the characterization of these potentially new resistance resources particularly regarding their durability and effectiveness at other locations and environments. Furthermore, the identification of R gene(s) or quantitative trait loci (QTLs) mediating disease resistance in these genotypes is important for marker-assisted breeding. It is not clear, how many different resistance traits are present in the different resistant genotypes discussed. It can be assumed that some of these cultivars have the same resistance mechanism. A pedigree analysis combined with the identification of common ancestors with similar resistance may help to define groups of cultivars, which carry potentially the same resistance trait. A large dataset describing a multi-generation pedigree in the apple germplasm calculated from whole-genome single nucleotide polymorphism (SNP) data is available [52], which includes about 1400 unique genotypes (*Malus* UNiQue genotypes—MUNQ, Denance et al. [53]) of different apple cultivars. Unfortunately, the pedigree of only a few of the genotypes identified in this work are included in this data set (data not shown), which does not allow any statement about common groups. Further SNP analyses, e.g., using the 20 k Illumina SNP array [54] or the 480 k Axiom SNP array [55] available for apple, would be necessary to analyze pedigrees and to trace back the origin of resistances.

## 5. Conclusions

Using a systematic screening of the apple cultivar collection of the Fruit Gene bank in Dresden-Pillnitz, several cultivars with high value for resistance breeding and sustainable cultivation were identified. Furthermore, our work provides a well-defined basis for the discovery of undescribed, new scab and powdery mildew resistances.

The information on the resistance/susceptibility of fruit genetic resources towards economically important diseases is important for breeding and for replanting traditional cultivars. Genetic diversity provides the raw material for breeding and plant improvement. It allows breeders to react to new arising requirements of consumers and markets as well as to climate change. Whereas in earlier years only phenotypic characteristics were evaluated, phenotypic and genotypic evaluation is now used in practice. The aim is to achieve a durable disease resistance in fruit breeding with modern varieties where resistance is based on multiple genes. For landscape conservation, growing apples under extensive conditions in meadows, it is important to use resistant or low susceptible old or new cultivars based on results of scientific evaluation.

Based on the assessment of scab and mildew resistance by field evaluation and molecular markers, a range of cultivars could be recommended for growing in extensive plantings scattered on pastures and along field tracks, where normally no fungicides are applied. Furthermore, these cultivars can be used as donors in breeding. Among them are modern varieties, such as Ariwa, Discovery, typical juice cultivars, like Börtlinger Weinapfel and Erbachhofer, but also old traditional varieties with reliable fruit characteristics, such as Altländer Pfannkuchenapfel, Rote Sternrenette, and Schöner aus Empt.

## Figures and Tables

**Figure 1 plants-10-01227-f001:**
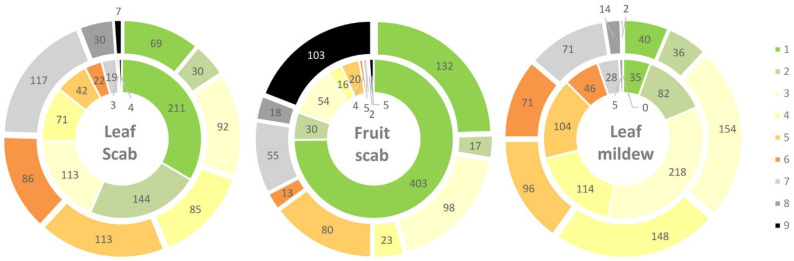
Proportion of cultivars with scab and powdery mildew incidence according to Appendix A (1 no symptoms–9 highest incidence) in 2012 (inner circle) and 2013 (outer circle).

**Table 1 plants-10-01227-t001:** Assessment scale for apple scab (*V. inaequalis*) on leaves, according to www.vinquest.ch (accessed on 10 May 2021), and fruits as well as for apple powdery mildew (*P. leucotricha*).

Scale	Susceptibility	Scab Leaf	Scab Fruit	Mildew
**1**	No infection	No visible macroscopic symptoms	No visible macroscopic symptoms	No visible macroscopic symptoms
**2**	Very low	A few small scab spots are detectable only visible on closer inspection	A few small scab spots are detectable only visible on closer inspection	1 sporolating spot
**3**	Low	Visible lesions, very thinly scattered in the tree	Visible lesions, very thinly scattered in the tree	up to 25% of leaves covered with infections symptoms
**4**	Low–Medium			>25% and <50% of leaves covered with infections symptoms
**5**	Medium	Numerous lesions widespread over a large part of the tree	Numerous lesions spread over a large part of the fruits	50% of leaves covered with infections symptoms
**6**	Medium–High			>50% and <75% of leaves covered with infections symptoms
**7**	High	Severe infection with half of the leaves infected by multiple lesions	Severe infection with half of the fruits infected by multiple lesions	75% of leaves covered with infections symptoms
**8**	High–Extremely high			>75% and <100% of leaves covered with infections symptoms
**9**	Extremely high	Tree completely affected with (nearly) all the leaves badly infected by multiple lesions	Tree completely affected with (nearly) all the fruits badly infected by multiple lesions	100% of leaves covered with infections symptoms

**Table 2 plants-10-01227-t002:** Cultivars with low susceptibility to scab (leaf and fruit). Apple cultivars of the JKI collection with scab rating scales 1 (no symptoms) and 2 (very low scab) in 2012 to 2013 are listed and additional data from experiments obtained in 1997 and 1999 and from 2006 to 2007 are supplemented. *R genes* indicated by the respective marker allele(s) from molecular analysis are given.

Cultivar	Period						*R* Genes Indicated by Respective Marker Alleles
	1997	1999 *	2006	2007 **	2012	2013	
	Scab Ratings On						
	Leaf	Fruit	Leaf	Fruit	Leaf	Fruit	
Ahrista	1	1	1	1	2	1	*Rvi6*
Akane	1	1	2	2	1	1	
Altländer Rosenapfel ^a^	1	1	1	1	1	1	*Rvi14*
Aneta	n.d.	n.d.	1	1	1	2	*Rvi6 Rvi13*
Angold	1	1	2	1	1	1	*Rvi17*
Antonovka	1	1	1	2	1	1	*Rvi14 Rvi17*
Antonovka Kamenička	2	2	1	2	1	1	*Rvi14 Rvi17*
Ariwa ^a^	n.d.	n.d.	1	1	1	1	*Rvi6*
Bessemânka Mičurinskaâ	1	1	1	1	2	1	*Rvi17*
Börtlinger Weinapfel	1	2	1	2	2	1	
Crimson Crisp	n.d.	n.d.	1	1	2	1	*Rvi6*
Dalinred	n.d.	n.d.	1	1	2	2	*Rvi6 Rvi13*
Discovery	1	2	1	1	1	1	
Dorheimer Streifling	n.d.	n.d.	n.d.	n.d.	1	1	
Ecolette	1	1	1	1	2	2	*Rvi6*
Engelshofer	1	1	1	1	1	1	
Englischer Prinz	1	1	2	1	1	1	*Rvi14*
Finkenwerder Prinzenapfel	3	2	1	1	2	1	
Franksenapfel	1	n.d.	1	1	1	1	
Früher Viktoria	2	1	2	2	2	1	*Rvi6*
Gaesdonker Renette ^a^	n.d.	n.d.	n.d.	n.d.	2	1	
Geflammter Kardinal	n.d.	n.d.	n.d.	n.d.	2	1	
Generos	n.d.	n.d.	1	1	1	1	*Rvi13*
Gochsheimer ^a^	n.d.	n.d.	n.d.	n.d.	2	1	
GoldRush	1	1	1	1	2	1	*Rvi6 Rvi13*
Grenadier	1	1	2	1	2	1	*Rvi6*
Gretapfel	n.d.	n.d.	n.d.	n.d.	1	1	*Rvi17*
Gustavs Dauerapfel	n.d.	n.d.	n.d.	n.d.	2	2	
Hadelner Sommerprinz	n.d.	n.d.	n.d.	n.d.	2	1	*Rvi14*
Heinemanns Schlotterapfel	n.d.	n.d.	n.d.	n.d.	1	1	
Hibernal 4n ^a^	1	1	2	1	1	1	*Rvi14 Rvi17*
Hochzeitsapfel	n.d.	n.d.	n.d.	n.d.	2	1	*Rvi14*
Jeverländer Süßapfel ^a^	n.d.	n.d.	n.d.	n.d.	1	1	*Rvi13*
Laxtons Fortune ^a^	2	1	1	1	2	1	
Linsenhofer Sämling ^a^	n.d.	n.d.	n.d.	n.d.	2	1	
Nela	1	1	1	1	1	1	*Rvi6*
Oberöstereich. Brünnerling	4	1	3	1	2	1	
Odenwälder	1	1	2	2	2	2	*Rvi6*
Oetwiler Renette	n.d.	n.d.	n.d.	n.d.	2	1	
Porzenapfel ^a^	n.d.	n.d.	1	1	2	1	
Primula	1	1	2	1	2	1	*Rvi6*
Prinz Albrecht von Preußen	4	1	1	3	2	1	
Purpurroter Cousinot	4	1	n.d.	n.d.	1	1	*Rvi6*
Purpurroter Zwiebelapfel ^a^	n.d.	n.d.	n.d.	n.d.	2	1	
Realka ^a^	1	1	1	2	1	1	*Rvi2 Rvi4 Rvi6*
Red Boy	n.d.	n.d.	n.d.	n.d.	2	1	
Red Topaz	n.d.	n.d.	n.d.	n.d.	1	1	*Rvi6*
Regunde	1	1	1	1	2	2	*Rvi6*
Reka	1	1	1	1	1	1	*Rvi2*
Relinda	n.d.	n.d.	n.d.	n.d.	2	2	*Rvi6*
Remura	1	1	1	1	1	1	*Rvi4*
Rene	n.d.	n.d.	3	2	1	1	*Rvi6*
Retina	1	1	3	1	2	1	*Rvi6 Rvi13*
Rheinischer Winterrambur	n.d.	n.d.	1	4	2	1	
Ritters Stolz	n.d.	n.d.	1	3	2	1	
Schöner aus Elmpt ^a^	n.d.	n.d.	n.d.	n.d.	1	1	*Rvi13*
Schöner aus Wiltshire	n.d.	n.d.	3	1	1	1	
Steinbacher	n.d.	n.d.	n.d.	n.d.	1	1	
Topaz	2	1	2	2	1	1	*Rvi6*
Welschisner	1	1	1	2	1	1	

* data obtained from the Institute for Plant Genetics and Research on Cultivated Plants in the Fruit Gene bank Dresden the predecessor of the today’s institution and partly published [15,16]; ** results obtained from scab susceptibility rating in the same plot at JKI in 2006 and 2007 like in 1997 and 1999; ^a^ cultivars with low susceptibility to both diseases, scab and powdery mildew; n.d. not determined.

**Table 3 plants-10-01227-t003:** Cultivars with low susceptibility to powdery mildew. Apple cultivars of the JKI collection with susceptibility rating scales 1 (no symptoms) and 2 (1 sporolating spot) in 2012 to 2013 are listed. Additional data from experiments obtained in 1997, 1999 and from 2006 to 2007 are supplemented. *R* genes indicated by the respective marker allele(s) from molecular analysis are given.

Cultivar	1997 *	1999 *	2006–2007 **	2012–2013	Average (Median)	*R* Genes Indicated by Respective Marker Alleles
Altländer Pfannkuchenapfel	2	5	5	2	3.5	
Altländer Rosenapfel ^a^	2	5	5	2	3.5	
Antonovka Polutorafuntovaâ	2	3	3	2	2.5	
Ariwa ^a^	n.d.	3	3	2	3	*Pl1*
Ashmeads Kernel	3	4	3	2	3	
Baumanns Renette	n.d.	n.d.	2	2	2	
Betzinger Grünapfel	n.d.	n.d.	n.d.	1	1 ***	
Böblinger Straßenapfel	n.d.	n.d.	n.d.	1	1 ***	
Bockenhusen	n.d.	n.d.	n.d.	2	2 ***	
Börtlinger Weinapfel ^a^	2	4	2	2	2	
Bramleys Seedling	3	3	4	2	3	*Pld*
Bratzelapfel	n.d.	n.d.	n.d.	2	2 ***	
Erbachhofer	2	2	1	2	2	*Plm*
Gaesdonker Renette ^a^	n.d.	n.d.	n.d.	1	1 ***	
Gochsheimer ^a^	n.d.	n.d.	n.d.	1	1 ***	
Göhrings Renette	n.d.	n.d.	n.d.	2	2 ***	
Großer Api	n.d.	n.d.	n.d.	2	2 ***	*Plm*
Hibernal 4n	2	3	3	2	2,5	
Jakob Fischer	2	4	2	2	2	
Jeverländer Süßapfel ^a^	n.d.	n.d.	n.d.	1	1 ***	
Juliane	n.d.	n.d.	n.d.	2	2 ***	
Kardinal Bea	2	2	2	2	2	
Krügers Dickstiel	4	4	2	2	3	
Laxtons Fortune ^a^	3	4	3	2	3	
Leistadter Rotapfel	n.d.	n.d.	n.d.	2	2 ***	
Linsenhofer Sämling ^a^	n.d.	n.d.	n.d.	2	2 ***	
Peasgoods Sondergleichen	1	2	3	2	2	
Pfaffenhofer Schmelzling	n.d.	n.d.	n.d.	2	2 ***	
Pomme d’Or	n.d.	n.d.	n.d.	1	1 ***	
Porzenapfel ^a^	n.d.	n.d.	3	2	2.5	
Präsident Decour	n.d.	n.d.	n.d.	2	2 ***	
Prima	n.d.	n.d.	n.d.	1	1 ***	
Realka ^a^	3	2	3	2	2,5	
Riesenboiken	2	3	2	2	2	
Roter Altländer Pfannkuchenapfel	4	5	3	2	3.5	
Roter Sossenheimer	n.d.	n.d.	n.d.	2	2 ***	
Schöner aus Elmpt ^a^	n.d.	n.d.	n.d.	2	2 ***	
Sonnenwirtsapfel	n.d.	n.d.	n.d.	1	1 ***	
Sulinger Grünling	n.d.	n.d.	n.d.	2	2 ***	
Virginia Crab	n.d.	n.d.	n.d.	1	1 ***	*Pl1*
Welschweinling	2	5	4	2	3	
Wildeshausener Goldrenette	1	3	7	1	2	

* data obtained from the Institute for Plant Genetics and Research on Cultivated Plants in the Fruit Gene bank Dresden the predecessor of the today’s institution and partly published [15,16]; ** results obtained from phenotyping powdery mildew in the same plot at JKI in 2006 and 2007 like in 1997 and 1999; *** phenotyping recorded only in 2012–2013; ^a^ cultivars with low susceptibility to both diseases, scab and powdery mildew; n.d. not determined.

**Table 4 plants-10-01227-t004:** Fragment sizes for selected markers linked to scab and powdery mildew resistance genes tested on a subset of the scab differential host set (http://www.vinquest.ch/monitoring/establishing_network.htm, accessed on 10 May 2021), complemented by four more genotypes. The first row per genotype represents the fragment sizes obtained in this study and the second line (grey) represents the respective reference alleles. Fragment sizes for all markers are given only for Gala and Golden Delicious, for all other genotypes fragment sizes are only given if the marker is linked to the respective resistance gene or if a genotype amplified a fragment of the same length. Alleles linked to a resistance genes are given in bold and all reference alleles are given in italics.

**Marker Linked to**	***Rvi2***					***Rvi4***		***Rvi15***								***Rvi6***						
			**Rvi11**		**Rvi8**	**Rvi15**		**Rvi4**			**Rvi5**					**Rvi17**		**Rvi8**	**Rvi11**			
**Genotypes (R–Genes)**	**CH02b10**		**CH05e03**		**OPL19SCAR ^a^**	**CH02c02a**		**CH02f06**		**Vr2C_UTR ^k^**	**Hi07h02**		**FMACH_Vm2 ^g^**	**FMACH_VM3 ^g^**		**CH-Vf1**		**OPB18SCAR ^b^**	**CH03d01**		**CH02c06**	
H(0) Gala	126	132	169	181	430	142	178	139	159		247	262	142	249		141		633	91	101	235	239
	*130 ^a^*	*136 ^a^*	*179 ^a^*	*191 ^a^*		*149 ^a^*	*185 ^a^*	*144 ^a^*	*165 ^a^*		*251 ^a^*	*267 ^a^*				*146 ^a^*						
H(1) Golden Delicious (*Rvi1*)	123	126	174	181	430	178	183	144	159		247	255	142	241	249	141	173	650	113		235	239
	*126 ^a^*	*130 ^a^*	*185 ^a^*	*191 ^a^*		*185 ^a^*	*191 ^a^*	*150 ^a^*	*165 ^a^*		*251 ^a^*	*259 ^a^*				*146 ^a^*	*180 ^a^*					
H(2) TSR34T15 (*Rvi2*)	**122**	146	**163**		**430**			147	150													
	***125 ^a^***	*152 ^a^*	***165 ^b^***		***430 ^a^***																	
H(4) TSR33T239 (*Rvi4*)					430	142	**176**	**147**	150	**522**												
						*149 ^a^*	***183 ^a^***															
H(5) 9-AR2T196 (*Rvi5*)					430						**226**	273	**154**	**355**								
											***230 ^a^***	*277 ^a^*	***158 ^g^***	***355 ^g^***								
H(6) Priscilla (*Rvi6*)					430			147	150							**159**	173					
																***166 ^a^***	*180 ^a^*					
H(8) B45 (*Rvi8*)					**430**													**755**				
					***430 ^a^***													***799 ^f^***				
H(11) *M. baccata* var. jackii (*Rvi11*)			**150**		430														103	105	233	245
			***160 ^a^***																*109 ^c^*	***115 ^c^***	*222 ^c^*	***248 ^c^***
H(12) Hansens *baccata* #2 (*Rvi12*)					430																	

H(13) Durello di Forli (*Rvi13*)					430																	

H(14) Dülmener Rosenapfel (*Rvi14*)					430																	

H(15) GMAL 2473 (*Rvi15*)	122	132			430	**176**	179	**147**	159	**522**						139	141					
						***183 ^a^***	*187 ^a^*	***152 ^a^***	*165 ^a^*													
04-214-79 (*Rvi17*)					430											**139**	173					
06_57 (*Pl1 Pl2 Plm*)					430																	

D12 (*Pld*)					430	144	176															

MIS (*Plm*, *Rvi16*)					430																	
**Marker Linked to**	***Rvi12***				***Rvi13***				***Rvi14***		***Rvi16***				***Pl1***		***Pl2***		***Pld***		***Plm***	
**Genotypes (R–Genes)**	**SSR-23.17 ^d^**		**SSR-24.91 ^d^**		**CH02b07**		**CH04f03**		**HB09 ^i^**		**NZmsCN943818**		**NH030 a**		**AT20Scar ^h^**		**Pl2_F1/R1 ^k^**		**CH03c02 ^j^**		**CH02d12**	
H(0) Gala	273	288	216	235	103	111	175	185	191	222	197		185	195	495				123	125	199	
	*271 ^d^*	*285 ^d^*	*217 ^d^*	*235 ^d^*	*112 ^a^*	*120 ^a^*	*181 ^a^*	*191 ^a^*	*197 ^a^*	*228 ^a^*												
H(1) Golden Delicious (*Rvi1*)	273	281	216	235	103	111	185		222	232	197		185	195	492	495			123	125	195	199
					*112 ^a^*	*120 ^a^*	*191 ^a^*	*197 ^a^*	*238 ^a^*													
H(2) TSR34T15 (*Rvi2*)							185	191														

H(4) TSR33T239 (*Rvi4*)							175	185														

H(5) 9-AR2T196 (*Rvi5*)							185	191														

H(6) Priscilla (*Rvi6*)							175	185														

H(8) B45 (*Rvi8*)							175	185														

H(11) M. *baccata* jackii (*Rvi11*)																						

H(12) Hansens *baccata* #2 (*Rvi12*)	**244**	274	200	***208***											450							
	***242 ^d^***	*272 ^d^*	*201 ^d^*	***209 ^d^***																		
H(13) Durello di Forli (*Rvi13*)					**111**	126	**185**	187													185	199
					***120 ^a^***	*134 ^a^*	***191 ^a^***	*193 ^a^*														
H(14) Dülmener Rosenapfel (*Rvi14*)							179	185	**210**	232												
									***216 ^a^***	*238 ^a^*												
H(15) GMAL 2473 (*Rvi15*)							175	185											102	129		

04-214-79 (*Rvi17*)					111	126	179	185														
06_57 (*Pl1 Pl2 Plm*)	244	281													**450**		**573**				**185**	205
															***450 ^h^***							
D12 (*Pld*)																			106	**129**		
																			125 ^j^	**131 ^j^**		
MIS (*Plm*, *Rvi16*)			199	208							**179**	197	185	**193**					120	125	177	**185**
											***198 ^e^***	216	202	***210 ^e^***					123 ^j^	127 ^j^	*197 ^e^*	***205 ^e^***

The respective fragment sizes are according to the following references: ^a^ Patocchi et al. [23], ^b^ Bus et al. [28], ^c^ Gygax et al. [25], ^d^ Padmarasu et al. [29], ^e^ Bus et al. [26], ^f^ Bus et al. [24], ^g^ Cova et al. [30], ^h^ Dunemann et al. [31], ^i^ Soufflet-Freslon et al. [32], ^j^ James et al. [27], ^k^ this study.

## Data Availability

Data is contained within the article or Appendix A.

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
