# Peer review of "Evaluation of Scab and Mildew Resistance in the Gene Bank Collection of Apples in Dresden-Pillnitz"

_plants, 2021, doi:10.3390/plants10061227_

Round 1

Reviewer 1 Report

The research deals with scab and powdery mildew resitsancies/suceptibilitis of a large set of traditional German apple cultivars. As such it is a very valuable type of mansucript emphasizing the need for screening available apple genetic resources for breeding and commercial purposes. Also the value of the genebank at Dresden-Pillnitz, Germany, becomes visible, although I suggest to point out a bit more the uniquness/short history of the collection. Overall, the paper reads well, English is perfect (as a non-native reviewer). A few things still need to be considered and I thus opt for major revisions.

General
- the results section contains citations (e.g. Bus, et al.) and interpretations (e.g. "Whether these cultivars represent new sources for scab resistance needs to be investigated in the future."). This is inappropriate and must be reserved for the discussion section!
- references are inconsistent (e.g. "&/and" Tree Genetics & Genomes vs. Genetic Resources and Crop Evolution; e.g. "capital letters/non capital letters" Tree Genetics & Genomes vs. Tree genetics & genomes; e.g. "abbreviations" Molecular Breeding vs Mol Breed; titles are sometimes capitalized/sometimes non-capitalized; species names are sometimes not italized)

Abstract:
- write out number 60 when beginning a sentence "60 cultivars" (also check throughout the text!)
- italize Malus floribunda

Materials and Methods
"[Ellis & Everh.]" = suggestion to use small caps for the authorities to avoid the use of different parenthesis/brackets. Also  ([Cooke]... Once stated here, no need to repeat the authority later on

The last paragraph of the introduction reads in parts like the abstract, the materials and methods section, and the conclusion, respectively, as authors state "will be....". It would be better to formulate hypothesis and aims to clearly indicate the goal of the present manuscript and to reformulate the sentences towards past tenses such as "was done to..." or "have been conducted to..." to link to the manuscript and not to some future work.

"Spindelbush" = "Spindlebush" OR "Spindelbusch"

"pomological characterization (step 1)": How was this done? Who was/how many people were involved?

Results
"scab ratings lower than 3 as well as those of powdery mildew ratings lower than 3 in both years" = "scab and powdery mildew ratings, respectively, lower than 3 in both years"

"Malus floribunda 821" Please provide authority

"M. baccata jackii", I suggest to either write "M. baccata var. jackii" or "M. baccata 'jackii'" "M. baccata 'Jackii'"

"(Qiagen, Hilden)" = "(Qiagen, Hilden, Germany)", at second appearance it is sufficient to write "(Qiagen)"

"(Thermo Fisher, Berlin)" = "(Thermo Fisher, Berlin, Germany)", also here second/third time only "(Thermo Fisher)

Also "(Sigma-Aldrich, Darmstadt)"...

"Nighty-four (14.9%)" ? = "Ninety-four (14.9%)"?

"501 and 389 cultivars" ff: Numbers on the beginning of a sentences should/must be written out.

"Bus, Bassett, Bowatte, Chagne, Ranatunga, Ulluwishewa, Wiedow and Gardiner [22]" = "Bus, et al. [22]"

Table 3
- why has the average used here, but not in Table 2?
- which average measure has been actually used, the arithmetic mean? If so, it is statistically wrong as the susceptibility as "rating scales" are of ordinal nature, which permits the use of the arithm. mean. Instead the median must be used

Table 4
- italizes species names (e.g. M. baccata)
- n.d. must be defined (although it is clear for most researchers)

"This needs to be included in consideration, as many location- and year-specific factors can have a great influence on the evaluation results. Year-to-year variation is often highly depending on climatic conditions, inoculum pressure and the growing condition of the planted trees." I wonder why this is stated although no weather data have been presented, at least for the years 2012 and 2013 although this could have been possible (even for the preceding years). Of course, as all cultivars are considered within the respective years, the expression of resistances/susceptibilities is somehow systematic and requires theoretically no information on weather data. Nonetheless, it is important to understand how the different resistances/susceptibilities are affected by weather conditions per year (as there seem to be huge difference - especially to scab) as well as to eventually derive potential managements strategies (pruning, exposure to sun,...). I just see that management implications have been stated in the last paragraph of the conclusion, but given the applicability of the results also for a large non-research community, I deem an emphasize on the practical application within the Discussion section is of high value. Here, it should be discussed if the management practices with relation to scab and mildew have been considered in earlier articles or if/how they are applied in the field already (the latter is certainly more likely?).

"was done according to [19]," = "was done according to AUTHOR [19],"

"Further SNP analyses would be necessary, that" = "Further SNP analyses would be necessary that"

The last paragraph of the discussion rather reads like the beginning of the conclusion. I suggest to shift it to there.

Table S1
- suggestion: merge the headers such as "Scab leaf 2012" to one "Scab leaf 2012 /2013"
- what means "Scale max." and why is it not presented for all cultivars?
- "n.d." must be defined (although it is clear for most researchers)
- "9 (very heavy infection). Data for each cultivar (2 trees) with the heaviest infection were considered" - mind the different font style here

Author Response

The comments are include in the pdf-file

Author Response

the Comments are include in the pdf-file

Round 2

Reviewer 1 Report

I would like to still suggest two changes

(1) Please add the authority for the first entry of Malus floribunda Siebold ex. Van Houtte clone 821

(2)  the results section contains citations (e.g. Bus, et al.) and interpretations (e.g. "Whether these cultivars represent new sources for scab resistance needs to be investigated in the future."). This is inappropriate and must be reserved for the discussion section!

Response:

These references are necessary for clarity of results in our opinion.

I understand your point, but formally it is against the the scientific writing standards (--> no references and interpretations in the results section!), which helps the reader to clearly distingush between author's work and third research findings

Author Response

Please see the file enclosed.
